# Pressure Transient Performance for a Horizontal Well Intercepted by Multiple Reorientation Fractures in a Tight Reservoir

**Guoqiang Xing [1,2], Shuhong Wu [1,2], Jiahang Wang [3], Mingxian Wang [1,4,*], Baohua Wang [1,2] and Jinjian Cao [1]**

[1] Research Institute of Petroleum Exploration and Development, China National Petroleum Corporation, Beijing 100083, China; m18600835372@163.com (G.X.); wush@petrochina.com.cn (S.W.); baohuawang@petrochina.com.cn (B.W.); caojinjian@petrochina.com.cn (J.C.)

[2] State Key Laboratory of Enhanced Oil Recovery, Ministry of Science and Technology, Beijing 100083, China

[3] Engineer Institute of SINOPEC Shanghai Offshore Oil & Gas Company, Shanghai 200120, China; wangjiahang.shhy@sinopec.com

[4] School of Earth Science and Engineering, Xi'an Shiyou University, Xi'an 710065, China

[*] Correspondence: wmx1012683002@163.com

**Abstract:** A fractured horizontal well is an effective technology to obtain hydrocarbons from tight reservoirs. In this study, a new semi-analytical model for a horizontal well intercepted by multiple finite-conductivity reorientation fractures was developed in an anisotropic rectangular tight reservoir. Firstly, to establish the flow equation of the reorientation fracture, all reorientation fractures were discretized by combining the nodal analysis technique and the fracture-wing method. Secondly, through coupling the reservoir solution and reorientation fracture solution, a semi-analytical solution for multiple reorientation fractures along a horizontal well was derived in the Laplace domain, and its accuracy was also verified. Thirdly, typical flow regimes were identified on the transient-pressure curves. Finally, dimensionless pressure and pressure derivative curves were obtained to analyze the effect of key parameters on the flow behavior, including fracture angle, permeability anisotropy, fracture conductivity, fracture spacing, fracture number, and fracture configuration. Results show that, for an anisotropic rectangular tight reservoir, horizontal wells should be deployed parallel to the direction of principal permeability and fracture reorientation should be controlled to extend along the direction of minimum permeability. Meanwhile, the optimal fracture number should be considered for economic production and the fracture spacing should be optimized to reduce the flow interferences between reorientation fractures.

**Keywords:** semi-analytical model; reorientation fractures; horizontal well; tight reservoir; flow behavior

## 1. Introduction

As one of the most effective stimulation techniques to increase the recovery of hydrocarbons, fractured horizontal wells have been widely adopted to develop tight reservoirs. In some tight reservoirs, reorientation fractures may be formed when an increase in pore pressure in a hydraulically-fractured area or the depletion of the reservoir pressure occurs [1,2]. For reorientation fractures, previous studies mainly focused on field case studies and fracture propagation [2,3]. Compared with hydraulically-fractured vertical wells, fractured horizontal wells can significantly enhance well productivity by increasing the reservoir contact area. For reservoir engineers, pressure transient analysis is one of the most effective techniques to diagnose the flow behavior of hydraulically-fractured

horizontal wells and evaluate the influence of key parameters on wellbore pressure, such as fracture spacing, fracture number and fracture configuration, which are intuitively important for efficient and economical development of tight reservoirs.

Research on the performance of horizontal wells with multiple hydraulic fractures has mainly focused on analytical and semi-analytical models [4–10]. Larsen and Herge presented the pressure transient behavior of horizontal wells with single or multiple finite-conductivity fractures in a three-dimensional unbounded reservoir using the Laplace transform method [6,7]. However, they only demonstrated the early-and middle-stage characteristics of multiple fractured horizontal wells without considering the outer boundary. Chen and Raghavan (1997) presented the solution of a multiply-fractured horizontal well in a rectangular drainage region by reforming the point-source solution of Ozkan and Raghavan [5], which laid a solid foundation for this study [8]. Zerzar and Bettam further extended our ability to understand the pressure behavior of a horizontal well with several fractures in closed anisotropic reservoirs [10]. However, most previous models are introduced based on the assumption that the hydraulic fracture is an ideal planar fracture. The direct observation on fractured cores and micro-seismic fracture monitoring demonstrated that the hydraulic fracturing technology can form complex fracture patterns [11–14], such as the non-planar fracture, the reorientation fracture. Thus, the planar fracture solution may not be accurate to analyze the pressure transient of horizontal wells with multiple reorientation fractures.

Recently, in order to analyze the flow behavior of non-planar fractures in vertical wells or horizontal wells, Luo et al. proposed that each fracture-wing should has its own coordinate system and fracture discrete equation to couple with the analytical reservoir solution, which was later called the fracture-wing method [15–18]. However, the fracture-wing method is based on a harsh condition that the exchange of the fluid only occurs in the junction of two fracture wings, which suggests that the fracture-wing model is also not applicable to horizontal wells with multiple reorientation fractures. Lately, by combining an analytical reservoir solution with a discretized fracture panel solution, the nodal analysis technique, which can calculate the pressure and flow rate of each discrete fracture node, was proposed to simulate the flow behavior for the well with complex fracture networks [19–21]. Although this technique may be suitable to analyze the flow behavior of fractured horizontal wells, this semi-analytical approach forms a large sparse matrix which may reduce the computational speed. In terms of the flow behavior of the reorientation fracture on a vertical well, Wu et al. [22] further employed the nodal analysis technique to establish a semi-analytical solution. Similarly, many discrete nodes will be definitely created when the nodal analysis technique is applied to the multiple fractured horizontal well model, which can increase the bandwidth of the related matrix and, thus, is not conductive to the rapid simulation of the multiple reorientation fractures.

To the best of our knowledge, most fractured horizontal well models are established based on the ideal planar fracture model. However, with the rapid development of hydraulic fracturing, reorientation fractures can be formed during the fracturing process, particularly in soft and shallow tight reservoirs [1,11,12]. For these complex multi-stage horizontal wells, the traditional fractured horizontal well model cannot easily describe the geometry of fracture patterns and analyze the transient flow behavior for fractured horizontal wells.

In this study, based on the combination of the nodal analysis technique and the fracture-wing method, we present a novel equation to describe the flow inside the reorientation fracture. Compared with the work of Wu et al. [22], the new fracture equation in this work can help to avoid calculating the pressure and flow rate at all fracture segments and, thus, can greatly improve the computational speed. On the basis of the new semi-analytical approach, we obtain a semi-analytical pressure solution for multiple reorientation fractures along a horizontal well in an anisotropic tight reservoir.

The remainder of this paper is organized as follows: In Section 2, we present the basic physical description of the model. The new semi-analytical solution based on the new fracture equation will be generated in Section 3. In order to verify the accuracy of the semi-analytical solution, we will use two special cases presented by the previous literature to compare with our semi-analytical

solution in Section 4. In Section 5, the flow characteristics and production rate distribution along each reorientation fracture will be discussed in detail. In addition, the influence of key parameters on pressure transient will also be investigated in Section 5, such as fracture number, fracture spacing and fracture configuration.

## 2. Physical Model

In this section, the physical model is introduced for pressure transient analysis of a horizontal well with multiple reorientation fractures in an anisotropic rectangular tight reservoir. Figure 1 presents the schematic of a horizontal well with two reorientation fractures, and every hydraulic fracture reoriented twice, which is significantly different from the physical model of Wu et al. [22]. Three parts, including principle fracture, irregular curve fracture and reoriented fracture, constitute the reorientation fracture. The basic assumptions are made as follows:

(1)　The tight reservoir is rectangular in shape. Single-phase Darcy flow occurs in the anisotropic tight reservoir. The permeability of the reservoir in the *x*- and *y*-direction is $k_x$ and $k_y$, respectively.

(2)　The horizontal well is fully intercepted by arbitrary reorientation fractures with constant height and width. The horizontal wellbore is deployed parallel to the *x*-axis, and all fractures' height is equal to the reservoir thickness.

(3)　Fluid in the horizontal wellbore and reservoir is constant viscosity and slightly compressible, and flow rates from each reorientation fracture contribute to the horizontal well total rate, although they may change over time. The horizontal wellbore is infinite-conductivity and no pressure loss occurs along the wellbore.

(4)　This reservoir is fully penetrated by all reorientation fractures. For the *i*-th reorientation fracture, its principal fracture angle is $\theta_{i,1}$, and its reoriented fractures angles are $\theta_{i,2}$ and $\theta_{i,3}$, respectively. All reorientation fractures have a finite conductivity and their tips are assumed to be impermeable boundaries.

(5)　Gravity effect is negligible, as well as the influence of the temperature on different reservoir parameters.

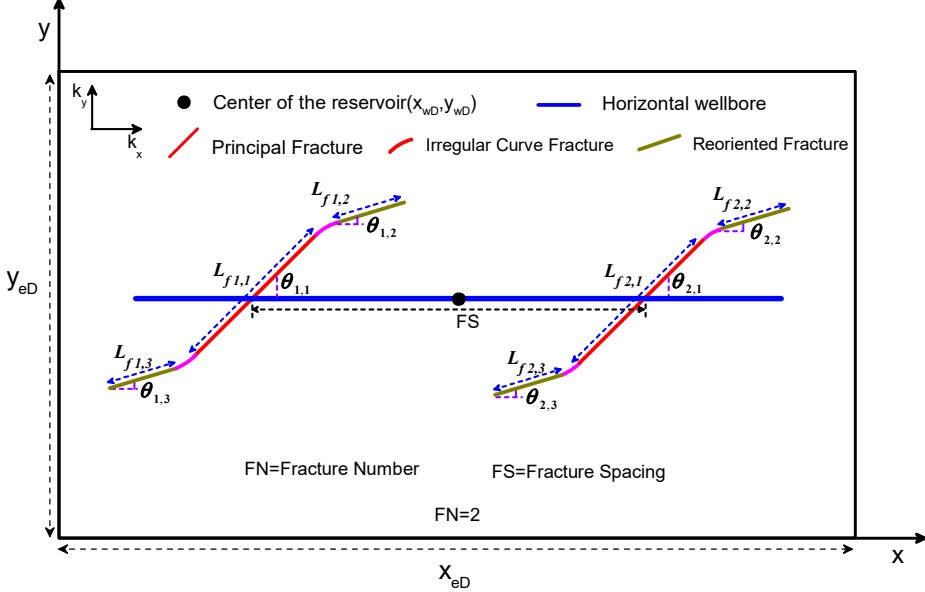

**Figure 1.** Schematic of two reorientation fractures along a horizontal well.

## 3. Mathematical Models

### 3.1. Dimensionless Definitions

To simplify the governing equation and definite conditions, the following dimensionless parameters are defined for the mathematical model.

The dimensionless pressure in the tight reservoir and all reorientation fractures, and the dimensionless time, are expressed as follows:

$$p_D = \frac{2\pi \bar{k} h (p_i - p)}{q_{sc} \mu B} \tag{1}$$

$$p_{fD} = \frac{2\pi \bar{k} h (p_i - p_f)}{q_{sc} \mu B} \tag{2}$$

$$t_D = \frac{\bar{k} t}{\mu (\phi c_t)_m L_R^2} \tag{3}$$

In the fracture model, the dimensionless fracture conductivity, $F_{cD}$, the dimensionless flow rate, $q_{fD}$, and the flow rate at the center of each fracture segment, $q_{fwD}$, are defined as follows:

$$F_{cD} = \frac{k_f w_f}{\bar{k} L_R}, \; q_{fD} = \frac{q_f L_R}{q_{sc}}, \; q_{fwD} = \frac{q_{fw}}{q_{sc}} \tag{4}$$

Other dimensionless definitions in the reservoir model or the reorientation fracture model are given as follows:

$$\beta = \sqrt{\frac{k_x}{k_y}}, \; x_D = \frac{x}{L_R}, \; y_D = \frac{y}{L_R}, \; l_D = \frac{l}{L_R}, \; \bar{k} = \sqrt{k_x k_y}, \; L_R = \left[ \sum_{i=1}^{N} \left( L_{fi,1} + L_{fi,2} + L_{fi,3} \right) + \sum_{i=1}^{m_i} L_i \right] / 2N \tag{5}$$

where $\beta$ is the anisotropic factor and $L_R$ is the reference length and $\bar{k}$ is the equivalent system permeability, $m_i$ is the total discrete number of all irregular curve fractures and $L_i$ is the length of the equivalent *i*-th planar fracture for the irregular curve fracture.

### 3.2. Reservoir Flow Model

On the basis of the Laplace transform, a fracture segment solution can be obtained for an anisotropic rectangular tight reservoir as follows [22]:

$$\tilde{p}_D = \frac{\pi \beta}{x_{eD}} \int_{l_D} \tilde{q}_{Di} \frac{\cosh \psi_0 \left( y_{eD} - |y_D \pm y_{Dmi}| \right)}{\psi_0 \sinh \psi_0 y_{eD}} dl_D$$

$$+ \frac{2\pi \beta}{x_{eD}} \int_{l_D} \tilde{q}_{Di} \sum_{n=1}^{\infty} \frac{\cosh \psi_n \left( y_{eD} - |y_D \pm y_{Dmi}| \right)}{\psi_n \sinh \psi_n y_{eD}} \cos\left( \frac{n\pi x_D}{x_{eD}} \right) \cos\left( \frac{n\pi x_{Dmi}}{x_{eD}} \right) dl_D \tag{6}$$

$$\psi_n = \sqrt{\left( s\beta + \left( \beta \frac{n\pi}{x_{eD}} \right)^2 \right)} \; n = 0, 1, 2 \cdots$$

where $(x_{Dmi}, y_{Dmi})$ is a movable infinitesimal unit along the fracture segment and $(x_D, y_D)$ is any point in the fracture segment.

However, Equation (6) has an implicit assumption that the minute fracture segment should be parallel to the coordinate axis. In order to get the pressure solution of a minute fracture segment with a reorientation angle $\theta$ (Figure 2) between the fracture and the horizontal wellbore, the following equations are used:

$$x_{Dmi} = x_{Di} + \chi \cos \theta_i \; y_{Dmi} = y_{Di} + \chi \sin \theta_i \tag{7}$$

$$\tilde{p}_{Di}(x_D, y_D, x_{Di}, y_{Di}, s) = \frac{\pi\beta}{x_{eD}} \sum_{i=1}^{N_I} \tilde{q}_{D,i} \int_{-\frac{\Delta l_{Di}}{2}}^{\frac{\Delta l_{Di}}{2}} \frac{\cosh\psi_0\left(y_{eD}-|y_D\pm(y_{Di}+\chi\sin\theta_i)|\right)}{\psi_0\sinh\psi_0 y_{eD}}d\chi$$

$$\frac{2\pi\beta}{x_{eD}} \sum_{i=1}^{N_I} \tilde{q}_{D,i} \int_{-\frac{\Delta l_{Di}}{2}}^{\frac{\Delta l_{Di}}{2}} \sum_{n=1}^{\infty} \frac{\cosh\psi_n\left(y_{eD}-|y_D\pm(y_{Di}+\chi\sin\theta_i)|\right)}{\psi_n\sinh\psi_n y_{eD}} \cos\left(\frac{n\pi x_D}{x_{eD}}\right)\cos\left(\frac{n\pi(x_{Di}+\chi\cos\theta_i)}{x_{eD}}\right)d\chi \tag{8}$$

$$\psi_n = \sqrt{s\beta+\left(\beta\frac{n\pi}{x_{eD}}\right)^2}\; n = 0, 1, 2\cdots; \; i = 1, 2\cdots N_I$$

where $(x_{Di}, y_{Di})$ is the midpoint of the fracture segment, $\theta_i$ represents the angle of $i$-th fracture segment, and $\triangle l_{Di}$ represents the dimensionless length of $i$-th reorientation fracture segment.

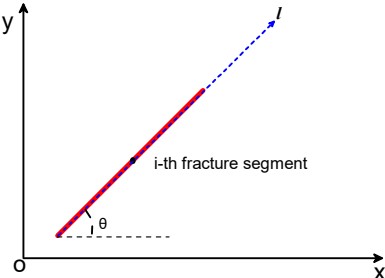

**Figure 2.** Fracture segment with an azimuth angle of $\theta$.

As stated previously, the reorientation fracture cannot be described by the ideal planar fracture. In order to stimulate the flow characteristics of a horizontal well intercepted by multiple reorientation fractures, we used the approximate approach similar to the boundary element method to handle the irregular shaped boundaries [23]. As illustrated in Figure 3, the irregular curve fracture is replaced by a series of planar fracture segments. Thus, the reorientation fracture system is considered to be composed of several planar fracture segments. The number of planar fracture segments used to describe the irregular curve fracture largely depends on the seepage of the irregular curve fracture.

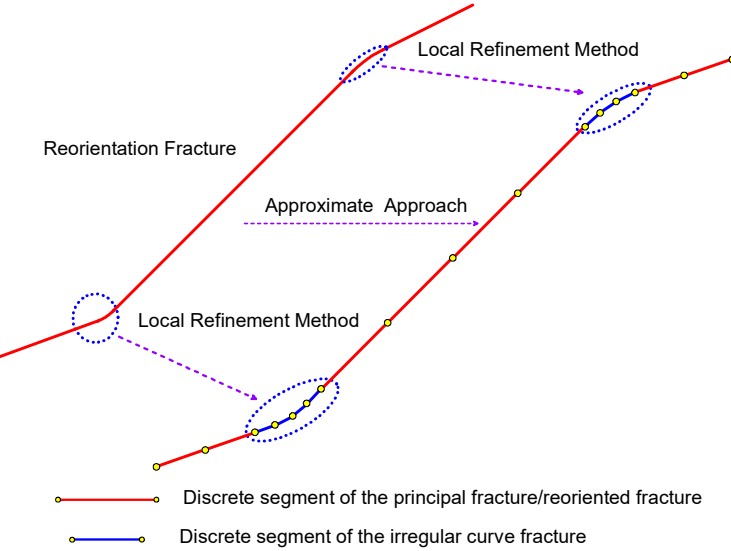

**Figure 3.** Approximate approach to handle the reorientation fracture.

### 3.3. Reorientation Fracture Flow Model

In this work, we assumed that the flow in the reorientation fracture is one-dimensional. Taking the $i$-th fracture segment as an example, we assume that the fluid flows from the $(i+1)$-th fracture

segment to the *i*-th fracture segment (Figure 4). Based on the results of Zhou et al. [19], the pressure difference between the *i*-th fracture segment and (*i*+1)-th fracture segment can be written as:

$$p_{fi+1} - p_{fi} = \int_{l_i}^{l_{i+0.5}} \frac{\mu B}{k_f w_f h} q_{fwi}(l,t)dl + \int_{l_{i+0.5}}^{l_{i+1}} \frac{\mu B}{k_f w_f h} q_{fwi+1}(l,t)dl \tag{9}$$

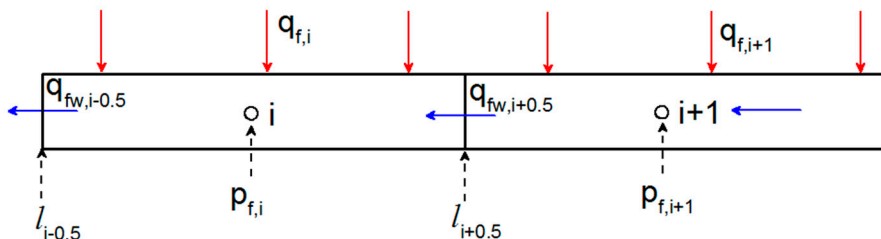

**Figure 4.** Schematic of fracture discretization.

In order to obtain the wellbore pressure, we take the integration method from the first fracture segment near the horizontal wellbore to the *i*-th fracture segment and derive Equation (10):

$$p_{fi} - p_w = \int_{l_{1-0.5}}^{l_{1+0.5}} \left(\frac{\mu B}{k_f w_f h}\right)_1 q_{fw1}(l,t)dl + \cdots + \int_{l_{(i-1)-0.5}}^{l_{(i-1)+0.5}} \left(\frac{\mu B}{k_f w_f h}\right)_{i-1} q_{fw(i-1)}(l,t)dl$$
$$+ \int_{l_{(i-1)+0.5}}^{l_i} \left(\frac{\mu B}{k_f w_f h}\right)_i q_{fwi}(l,t)dl \tag{10}$$

In addition, according to the principle of mass balance, we can obtain the following equations for each fracture segment:

$$q_{fwi} = q_{fw(i-0.5)} - q_{fi}(l - l_{i-0.5}) \qquad l_{i-0.5} \le l \le l_{i+0.5} \tag{11}$$

The inflow and outflow of each fracture node should satisfy the mass balance:

$$q_{fw,i-0.5} = \sum_{j=1}^{N_I} \left(q_{fj}\Delta l_j\right) - \sum_{j=1}^{i-1} \left(q_{fj}\Delta l_j\right) \tag{12}$$

where $N_I$ is total fracture segments of a wing of the *i*-th fracture.

Substituting Equation (11) and Equation (12) into Equation (10) yields:

$$p_{fi} - p_w = \int_{l_{1-0.5}}^{l_{1+0.5}} \left(\frac{\mu B}{k_f w_f h}\right)_1 \left[\sum_{j=1}^{N_I}\left(q_{fj}\Delta l_j\right) - q_{f1}(l - l_{1-0.5})\right]dl + \cdots +$$
$$\int_{l_{(i-1)-0.5}}^{l_{(i-1)+0.5}} \left(\frac{\mu B}{k_f w_f h}\right)_{i-1} \left[\sum_{j=1}^{N_I}\left(q_{fj}\Delta l_j\right) - \sum_{j=1}^{i-2}\left(q_{fj}\Delta l_j\right) - q_{fi-1}(l - l_{i-1-0.5})\right]dl +$$
$$\int_{l_{(i-1)+0.5}}^{l_i} \left(\frac{\mu B}{k_f w_f h}\right)_i \left[\sum_{j=1}^{N_I}\left(q_{fj}\Delta l_j\right) - \sum_{j=1}^{i-1}\left(q_{fj}\Delta l_j\right) - q_{fi}(l - l_{i-0.5})\right]dl \tag{13}$$

Based on above dimensionless definitions, Equation (13) can be rewritten as follows:

$$p_{wD} - p_{fDi} = \frac{2\pi}{F_{cD}}\left[l_{Di}\sum_{j=1}^{N_I}\left(q_{fDj}\Delta l_{Dj}\right) - \frac{\Delta l_{Di}^2}{8}q_{fDi} - \sum_{j=1}^{i-1}\left(\frac{\Delta l_{Dj}}{2} + l_{Di} - \sum_{n=1}^{j}\Delta l_{Dn}\right)\Delta l_{Dj}q_{fDj}\right] \tag{14}$$

To the best of our knowledge, Equation (14) is a new fracture equation and can be employed to describe the flow inside the reorientation fracture.

In order to obtain the fracture solution in Laplace domain, we should apply the Laplace transform to Equation (14) based on $t_D$ and thus we can have:

$$\tilde{p}_{wD} - \tilde{p}_{fDi} = \frac{2\pi}{F_{cD}} \left[ l_{Di} \sum_{j=1}^{N_I} \left( \tilde{q}_{fDj} \Delta l_{Dj} \right) - \frac{\Delta l_{Di}^2}{8} \tilde{q}_{fDi} - \sum_{j=1}^{i-1} \left( \frac{\Delta l_{Dj}}{2} + l_{Di} - \sum_{n=1}^{j} \Delta l_{Dn} \right) \Delta l_{Dj} \tilde{q}_{fDj} \right] \quad (15)$$

For the uniform dimensionless length of $\triangle l_{Di}$ for each fracture segment, Equation (15) will have the same form as that reported by Cinco-Ley et al., [24]. The simplification of Equation (15) is presented in Appendix A.

The flow equation of the reorientation fracture (Equation (15)) can be given in the matrix form:

$$\mathbf{C\tilde{q}_{fD} + D\tilde{p}_{fD} + E\tilde{p}_{wD} = 0} \quad (16)$$

where $\mathbf{\tilde{q}_{fD}}$ represents the vector of the flow-rate in the reorientation fracture, $\mathbf{\tilde{p}_{wD}}$ is the vector of the horizontal wellbore pressure and $\mathbf{\tilde{p}_{fD}}$ is the vector of the pressure in the reorientation fracture, C, D, and E are the corresponding coefficient matrices obtained from Equation (15).

### 3.4. Semi-Analytical Solutions

In this paper, we get the semi-analytical pressure solution by discretizing each reorientation fracture into a large number of planar fracture segments, similar to the method illustrated in Figure 4. On the basis of the superposition principle, we can easily get the dimensionless pressure solution for the *i*-th fracture segment in the Laplace domain, which is presented as a discretized form:

$$\tilde{p}_{Dk,i}\left(x_{Dk}, y_{Dk}, x_{Di}, y_{Di}, s\right) = \sum_{k=1}^{N} \sum_{i=1}^{N_I} \tilde{q}_{Dk,i} R_{Dk,i}\left(x_{Dk}, y_{Dk}, x_{Di}, y_{Di}, s\right) \quad (17)$$

where *N* is the reorientation fracture number and:

$$
\begin{aligned}
&R_{Dk,i}\left(x_{Dk}, y_{Dk}, x_{Di}, y_{Di}, s\right) = \\
&\frac{\pi\beta}{x_{eD}} \int_{-\frac{\Delta l_{Di}}{2}}^{\frac{\Delta l_{Di}}{2}} \frac{\cosh \psi_0 \left( y_{eD} - |y_{Dk} \pm (y_{Di} + \chi \sin\theta_i)| \right)}{\psi_0 \sinh \psi_0 y_{eD}} d\chi + \frac{2\pi\beta}{x_{eD}} \times \\
&\int_{-\frac{\Delta l_{Di}}{2}}^{\frac{\Delta l_{Di}}{2}} \sum_{n=1}^{\infty} \frac{\cosh \psi_n \left( y_{eD} - |y_{Dk} \pm (y_{Di} + \chi \sin\theta_i)| \right)}{\psi_n \sinh \psi_n y_{eD}} \cos\left(\frac{n\pi x_{Dk}}{x_{eD}}\right) \cos\left(\frac{n\pi (x_{Di} + \chi \cos\theta_i)}{x_{eD}}\right) d\chi \\
&\psi_n = \sqrt{s\beta + \left(\beta \frac{n\pi}{x_{eD}}\right)^2} \quad n = 0,1,2\cdots \quad i = 1,2\cdots N_I
\end{aligned}
\quad (18)
$$

Considering the continuity conditions of pressure and flux along the fracture surface, the following equations can be obtained in the Laplace domain:

$$\tilde{p}_D(x_D, y_D) = \tilde{p}_{fD}(x_D, y_D) \quad (19)$$

$$\tilde{q}_D(x_D, y_D) = \tilde{q}_{fD}(x_D, y_D) \quad (20)$$

Equations (17), (19), and (20) can be further written in the matrix form as follows:

$$\mathbf{A\tilde{q}_{fD} + B\tilde{p}_{fD} + O\tilde{p}_{wD} = 0} \quad (21)$$

where A and B are the corresponding coefficient matrix obtained from Equation (17).

Combining Equations (16) and (21) reveals the following simplified matrix:

$$\mathbf{G\tilde{q}_{fD}+H\tilde{p}_{wD}}=0 \tag{22}$$

where G and H are the corresponding coefficient matrices obtained from Equation (15) and Equation (17).

The sum of the flow rates of the reorientation fracture segments contributes to the total flow rate in the horizontal wellbore:

$$\sum_{k=1}^{N}\sum_{i=1}^{N_I}\left(\Delta l_{Dk,i}\tilde{q}_{fDk,i}\right)=\frac{1}{s} \tag{23}$$

Combining Equations (22) and (23), the following equation in the matrix form is obtained:

$$\begin{bmatrix} G & H \\ M^T & 0 \end{bmatrix}\begin{bmatrix} \tilde{q}_{fD} \\ \tilde{p}_{wD} \end{bmatrix}=\begin{bmatrix} 0 \\ \frac{1}{s} \end{bmatrix} \tag{24}$$

where $M^T$ is the vector of dimensionless length for each reorientation fracture segment.

$$M=\begin{bmatrix} \Delta l_{D1} & \Delta l_{D2} & \cdots & \Delta l_{Dsum} \end{bmatrix}^T \tag{25}$$

Equation (24) is the new semi-analytical solution for multiple reorientation fractures along a horizontal well in a tight reservoir in Laplace domain. According to Equation (24), the reference pressure of all fracture segments except the pressure of the horizontal wellbore does not need to be calculated and, thus, the calculation speed can be significantly increased, which is different from the work of Wu et al. [22]. By solving Equation (25) with the Gaussian elimination method, the pressure of the horizontal wellbore and the flow rate of each reorientation fracture segment can be obtained in Laplace space, and then can be further inverted into the real space by using the Stehfest numerical inverse method [25].

## 4. Verification and Comparison

The major advantage of the novel semi-analytical model is that we take the fracture reorientation and permeability anisotropy into consideration. To the best of our knowledge, there are no reports on a horizontal well perforated by a single reorientation fracture or multiple reorientation fractures in anisotropic rectangular reservoirs. Therefore, we conduct two special cases presented by Chen and Raghavan to verify the accuracy of our semi-analytical solution [8]. They calculated the pressure transient of a multiple fractured horizontal well in an isotropic rectangular reservoir using the reformulation of the point-source solution presented by Ozkan and Raghavan [4]. The comparison results are shown in Figure 5, where the semi-analytical solution shows a good fit to the results of Chen and Raghavan at every production time [8].

## 5. Results and Discussion

### 5.1. Transient Flow Characteristics

The transient flow behavior of this model is illustrated by transient-pressure type curves, which can be employed for pressure and rate transient analysis. In addition, some reservoir parameters can be obtained by type curves matching, such as the average reservoir permeability, fracture number, and fracture reorientation.

For a given set of parameters, the dimensionless pressure and its derivative of two reorientation fractures along the horizontal well in an anisotropic tight reservoir are displayed in Figure 6 to analyze its typical flow regimes. Figure 7 illustrates the fluid flow in the reservoir and reorientation fractures with different flow regimes. Without considering the transition flow regimes, we can easily observe six typical flow regimes in Figure 6: bilinear flow regime (BF), first linear flow regime (FLF),

first radial flow regime (FRF), second radial flow regime (SRF), pseudo-radial flow regime (PRF), and pseudo-steady-state flow regime (PSSF).

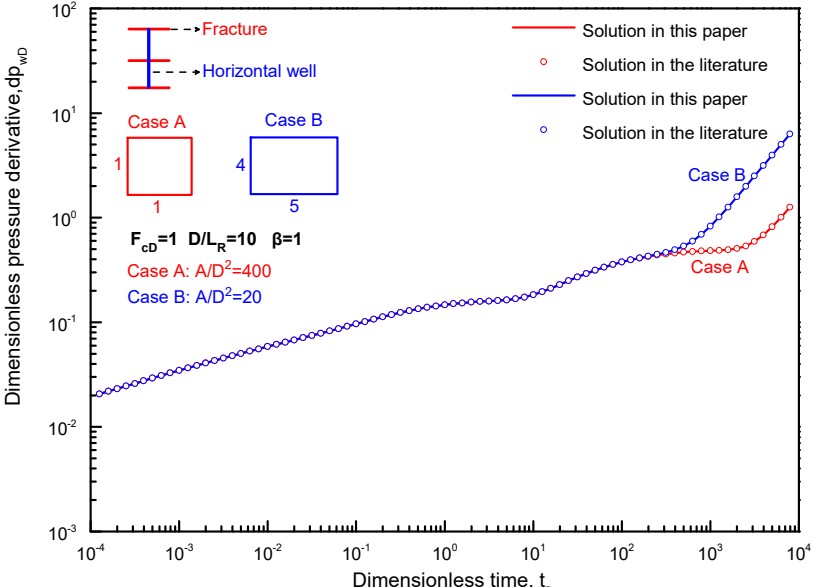

**Figure 5.** Comparison of pressure derivative for a horizontal well with three vertical fractures with the results of previous literature [8].

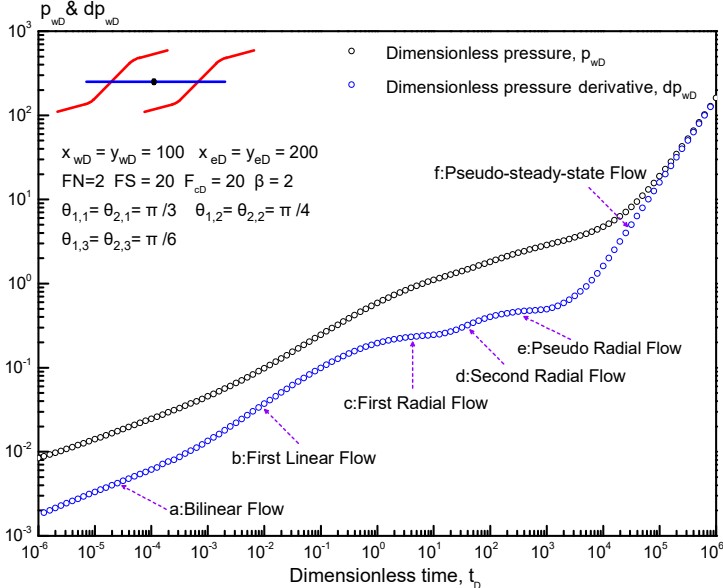

**Figure 6.** Typical curves of the pressure and pressure derivative for a horizontal well interpreted by two reorientation fractures.

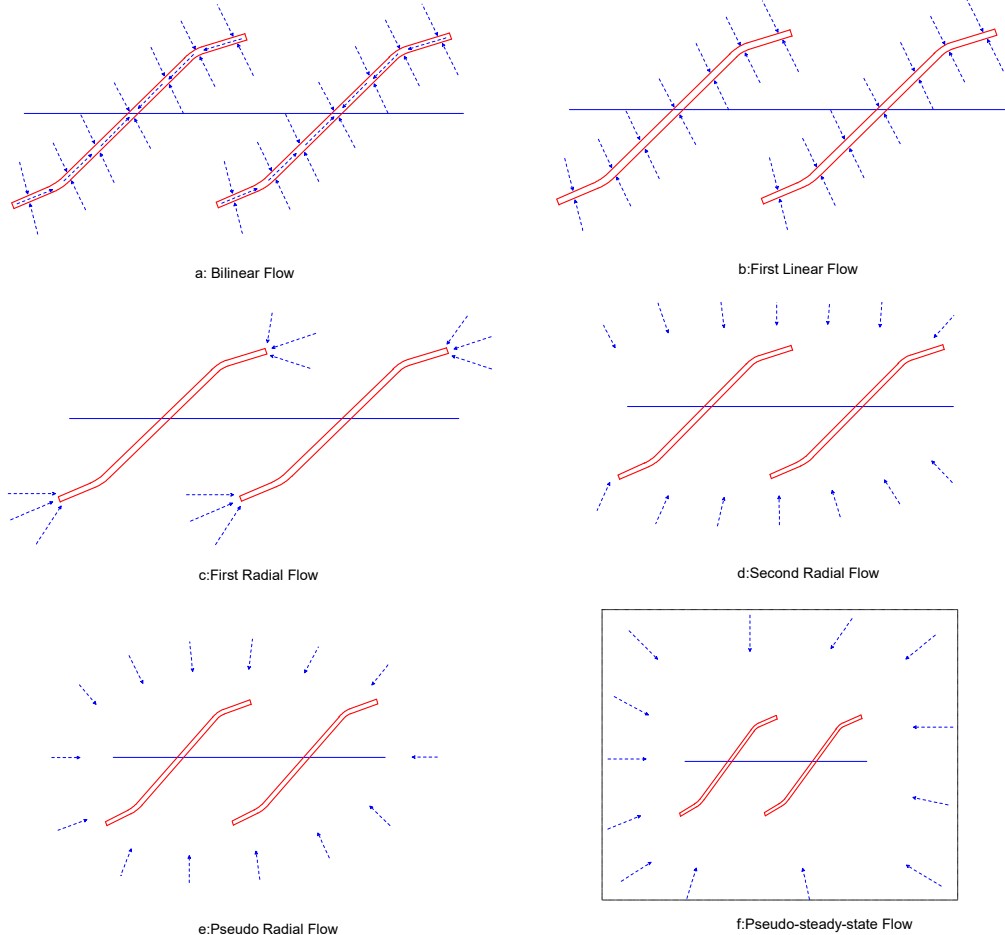

**Figure 7.** Schematic of the flow regimes for a horizontal well with two reorientation fractures: (**a**) Bilinear flow regime; (**b**) First linear flow regime; (**c**) First radial flow regime; (**d**) Second radial flow regime; (**e**) Pseudo radial flow regime; (**f**) Pseudo-steady-state flow regime.

Bilinear flow: In this regime, two types of linear flow, including the linear flow from the reservoir to reorientation fractures and from reorientation fractures to the horizontal wellbore, occur at the same time (Figure 7a). This flow behavior makes the slope of the dimensionless pressure derivative curve constant. In contrast to conventional multi-stage fractured horizontal wells, the streamline from the reservoir to the reorientation fracture may be not perpendicular to the fracture due to permeability anisotropy and fracture reorientation existing simultaneously. Consequently, the slope of the straight line at this regime is characterized by 1/4–1/3 in the log-log plot, not 1/4 as presented by Cinco-Ley et al. [24]. In addition, this flow behavior only occurs when the reorientation fracture conductivity is considered.

First linear flow: For the conventional finite-conductivity fractured horizontal well model, the slope of the pressure derivative curve is constant to 1/2 in this regime. Due to the fracture reorientation and permeability anisotropy, the slope in this regime is also larger than that of the conventional finite-conductivity fractured horizontal wells. During this regime, the fluid from the reservoir will linearly flow into the individual reorientation fracture and each reorientation fracture works independently on other reorientation fractures.

First radial flow: In the first radial flow regime, one of the most important transient flow characteristics is that the pressure derivative is expressed as a fixed value of 1/(2N). Consequently, the slope of the pressure derivative curve for various fractured horizontal well model may be different, which mainly depends on the fracture number. In this regime, the fluid around reorientation fractures will individually flow into the reorientation fractures. The duration of this regime is highly related to

adjacent fracture spacing and reorientation fracture length. Generally, a large reorientation fracture length or a wide fracture spacing can definitely lead to a long duration of the first radial flow regime.

Second radial flow: After the first radial flow, a second radial flow occurs in the tight reservoir. As demonstrated in Figure 6, the dimensionless pressure and its derivative curves are parallel to each other and the slope of these curves is a constant value of 0.36 in this regime. During this period, the interference of reorientation fractures starts to affect the flow between reorientation fractures and reservoir. After this regime, the transient behavior in the reservoir surrounded by the outer fractures reaches a pseudo-radial flow regime.

Pseudo radial flow: In this regime the dimensionless pressure derivative has a constant value of 0.5, and the derivative curve is parallel to the time axis (Figure 6). In the meantime, the fluid flow among the reservoir, the reorientation fractures, and the horizontal wellbore reach a dynamic balance and, thus, the dimensionless pressure derivative stabilizes at 0.5.

Pseudo-steady-state flow: A pseudo-steady-state flow regime occurs in the late-time period for all closed reservoirs. The typical characteristic of this regime is that the imensionless pressure and corresponding derivative curves increase rapidly and finally normalize to a straight line with a constant slope of 1. Generally, the seepage area in this flow regime always presents a circle (Figure 7f).

### 5.2. Production Rate Distribution

Figures 8 and 9 present the production rate distribution in each reorientation fracture for a horizontal well with three reorientation fractures and the data used in Figures 8 and 9 are presented in Table 1. In this section, we assumed that the inner reorientation fracture is fixed in the center of the tight reservoirs and the distance between the inner reorientation fracture and the outer reorientation fracture is assumed to be constant at FS = 20. Meanwhile, the horizontal wellbore extends parallel to the x-axis.

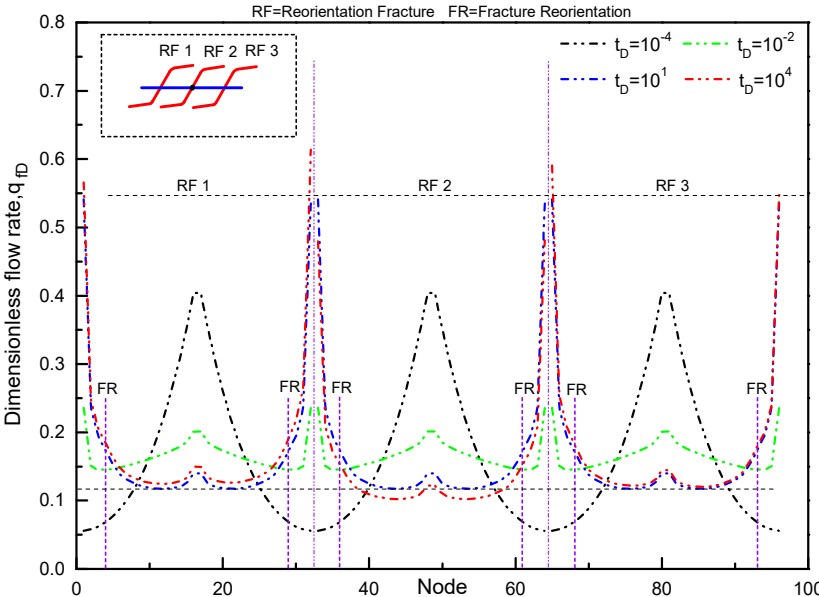

**Figure 8.** Rate distribution of a horizontal well with three reorientation fractures ($k_x = k_y$).

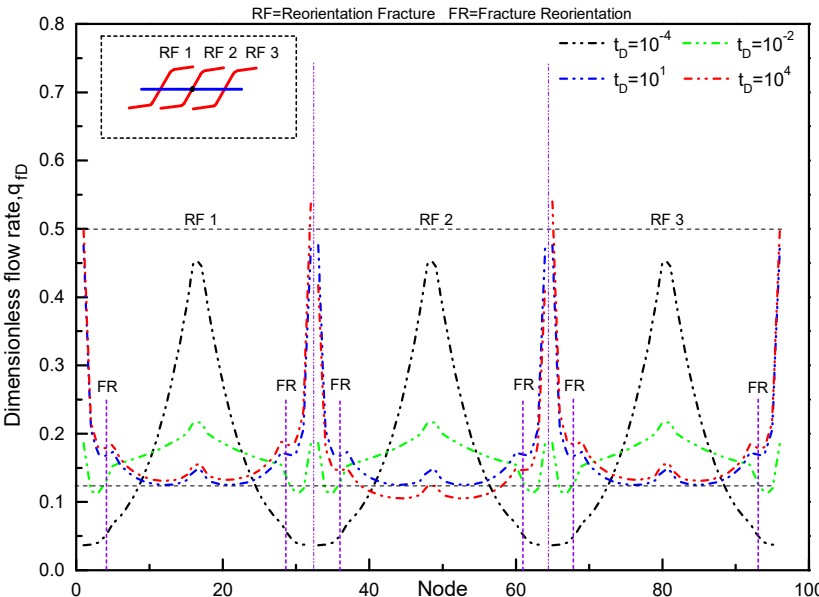

**Figure 9.** Rate distribution of a horizontal well with three reorientation fractures ($k_x = 4k_y$).

**Table 1.** Parameters used in Figures 8 and 9 for a horizontal well with three reorientation fractures.

| Basic Model Parameters | Value |
|---|---|
| Drainage area (dimensionless) | $200 \times 200$ |
| Fracture conductivity (dimensionless) | 20 |
| Principal fracture angle (deg) | $\theta_{1x} = \theta_{1y} = \theta_{1z} = 60°$ |
| Reoriented fracture angle (deg) | $\theta_{2x} = \theta_{2y} = \theta_{2z} = 30°$　$\theta_{3x} = \theta_{3y} = \theta_{3z} = 30°$ |
| Fracture spacing (dimensionless) | 20 |
| Inner fracture position (dimensionless) | $x_D = y_D = 100$ |

The production rate distribution of each reorientation fracture along the horizontal well in an isotropic tight reservoir is illustrated in Figure 8 when $k_x = k_y$. In the early flow regime (the black line in Figure 8), the production rate of each fracture segment is symmetrically distributed with respect to the wellbore node. As the pressure wave expands, the production rate in the reoriented fracture significantly increases along the extension of the reoriented fracture (the green line in Figure 8). For a long production time (Figure 8, $t_D = 10^4$), the production rate distribution changes significantly. Since the outermost fractures (RF1 and RF3) have a larger drainage area, they have a higher production rate. In addition, owing to the flow interference, the inner fracture (RF2) has a lower production rate. Furthermore, the production rate distribution of the outermost fractures (RF1 and RF3) (the red line in Figure 8) is asymmetrical. The reoriented fractures away from the horizontal wellbore have a larger production rate due to the larger seepage area caused by fracture reorientation.

Figure 9 presents the production rate distribution for each reorientation fracture in a horizontal well in an anisotropic tight reservoir ($k_x = 4k_y$, FN = 3). The production rate distribution within each reorientation fracture is similar to that in Figure 8. However, for an anisotropic reservoir, fracture reorientation has a more significant effect on the production rate in each reorientation fracture. In the reorientation section, the production rate distribution curves vary sharply (Figure 9), which suggests that the effect of fracture reorientation in anisotropic tight reservoir on type curves is significant and, thus, cannot be neglected.

### 5.3. Parameter Influence on Transient Pressure Behavior

In this section, we analyze the sensitivity of the transient pressure of multiple reorientation fractures along a horizontal well in an anisotropic tight reservoir. We consider the case of a horizontal well with two reorientation fractures to analyze the effects of some key parameters, such as principal

fracture angle (PFA), reoriented fracture angle (RFA), permeability anisotropic factor (PAF), and adjacent reorientation fractures spacing (FS), on the dimensionless pressure and its derivative curves. We also analyze the effect of reorientation fracture number and complex fracture configuration on type curves.

### 5.3.1. Effect of Principal Fracture Angle (PFA) on Type Curves

Figure 10 presents the effect of PFA on type curves for a horizontal well with two reorientation fractures. The dimensionless fracture spacing is set at 20. As displayed in Figure 10, PFA has a weak impact on type curves in the early-time period, i.e., the bilinear flow regime, first linear flow regime, and first radial flow regime. For an anisotropic tight reservoir, the dimensionless pressure and its derivative decrease as PFA increases, indicating that a large PFA is beneficial to improve the productivity of fractured horizontal wells. In addition, as PFA increases, the first radial flow regime occurs later. The effect of PFA on other flow regimes can be neglected.

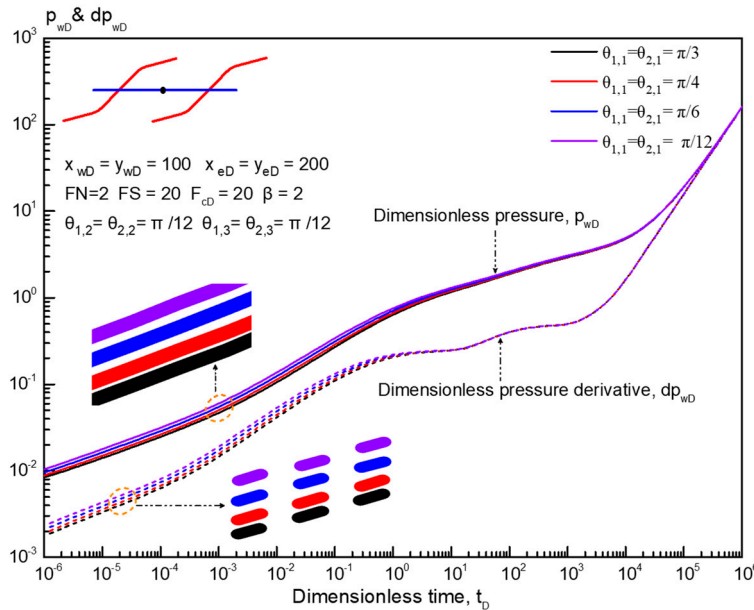

**Figure 10.** Effect of the principal fracture angle on pressure and the pressure derivative.

### 5.3.2. The effect of reoriented fracture angle (RFA) on type curves

Figure 11 shows the effect of RFA on the dimensionless pressure and pressure derivative curves. In order to completely reveal the effect of RFA on type curves, we compare the dimensionless pressure (PFA = RFA = $\pi/3$) with the dimensionless pressure (PFA $\neq$ RFA) and then find the point 'MRPC' (maximum relative pressure change) (Figure 12). The relative pressure change (RPC) is defined as follows:

$$RPC = \left(p_{wD|PFA\neq RFA} - p_{wD|PFA=RFA}\right)/p_{wD|PFA=RFA} \times 100\% \tag{26}$$

In Figure 12, $t_{DMRPC}$ represents the corresponding dimensionless time for the MRPC. Before $t_{DMRPC}$, as the RFA increases, the RPC decreases which means that RFA should be larger to maintain the constant rate under the same pressure depletion rate. However, after $t_{DMRPC}$, the effect of RFA on RPC weakens, which indicates that the effect of RFA on type curves tends to disappear. When RFA > PFA, RPC gets smaller and even becomes negative, which suggests that the larger the RFA (when RFA > PFA), the higher the well productivity.

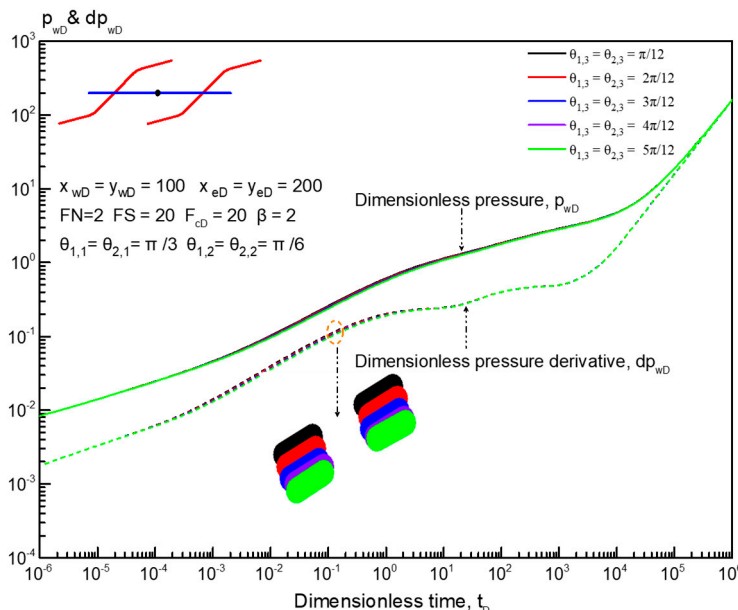

**Figure 11.** Effect of the reoriented fracture angle on pressure and the pressure derivative.

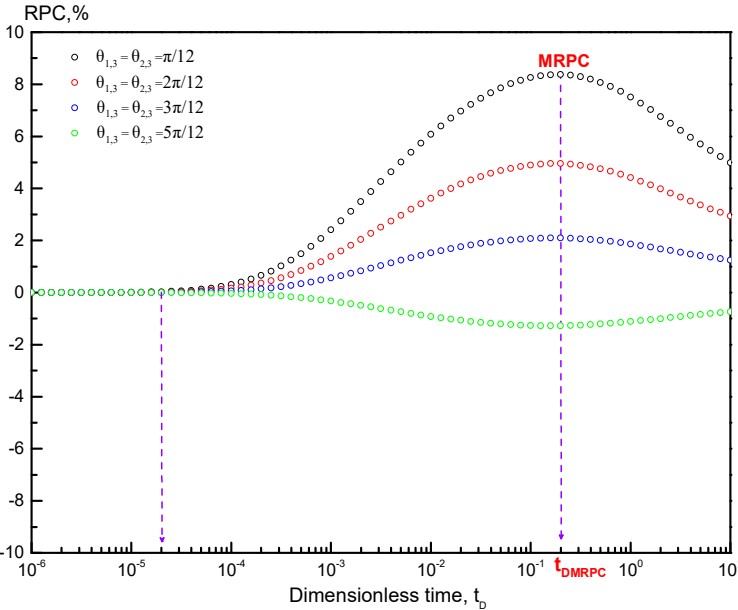

**Figure 12.** Reoriented fracture angle on relative pressure change.

### 5.3.3. Effect of Permeability Anisotropy on Type Curves

The impact of permeability anisotropy on dimensionless pressure and pressure derivative is illustrated in Figure 13 ($\beta > 1$) and Figure 14 ($\beta < 1$). In Figures 13 and 14, permeability anisotropy affects all the flow regimes except the pseudo-steady-state flow regime.

The influence of permeability anisotropy ($\beta \geq 1$) on type curves is displayed in Figure 13. Here, the horizontal wellbore is parallel to the principal permeability axis. As permeability anisotropic factor ($\beta$) increases, the dimensionless pressure and its derivative decrease in the early linear flow regimes, indicating that the pressure difference of the horizontal wellbore can be smaller to maintain the constant rate. However, before the pseudo-steady-state flow regime, the pressure and its derivative increase when the permeability anisotropic factor ($\beta$) increases, for the reason that the first radial flow regime tends to disappear when permeability anisotropic factor (PAF) increases to some extent.

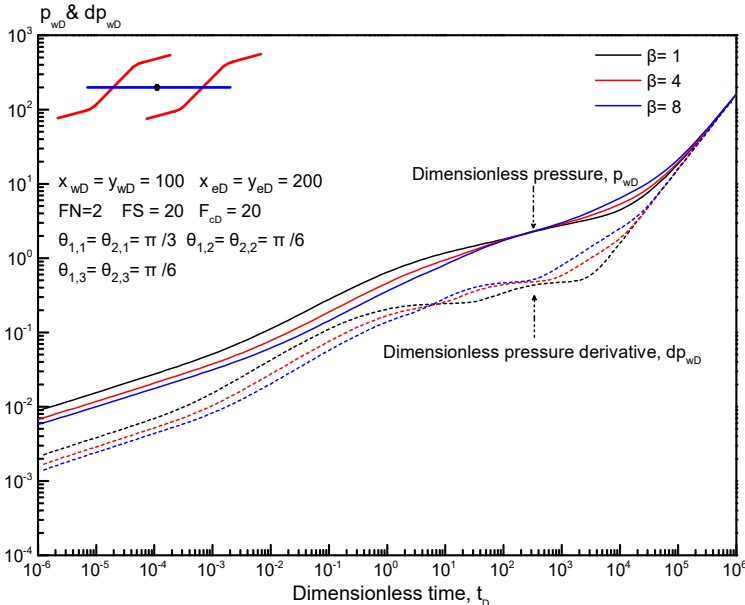

**Figure 13.** Effect of permeability anisotropy on pressure and the pressure derivative ($\beta \geq 1$).

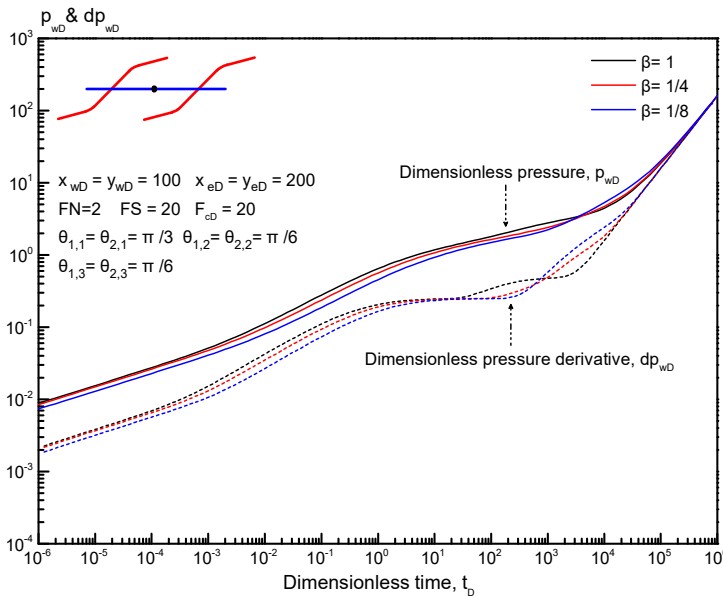

**Figure 14.** Effect of permeability anisotropy on pressure and the pressure derivative ($\beta \leq 1$).

Similarly, the effect of permeability anisotropic factor ($\beta \leq 1$) on type curves is shown in Figure 14. At this time, the horizontal wellbore is perpendicular to the principal permeability axis. Compared with Figure 13, we can find that the second radial flow regime tends to disappear when the horizontal wellbore is perpendicular to the principal permeability axis. Meanwhile, the compound linear flow regime tends to appear when the first radial flow regime ends, for the reason that the flow in the principal permeability direction dominates and, thus, the second radial flow regime is converted into the compound linear flow regime.

### 5.3.4. Effect of Fracture Conductivity on Type Curves

Figure 15 depicts the pressure and pressure derivative curves for a horizontal well intercepted by two reorientation fractures with various fracture conductivities: $\pi$, $10\pi$, $100\pi$, and $1000$, respectively.

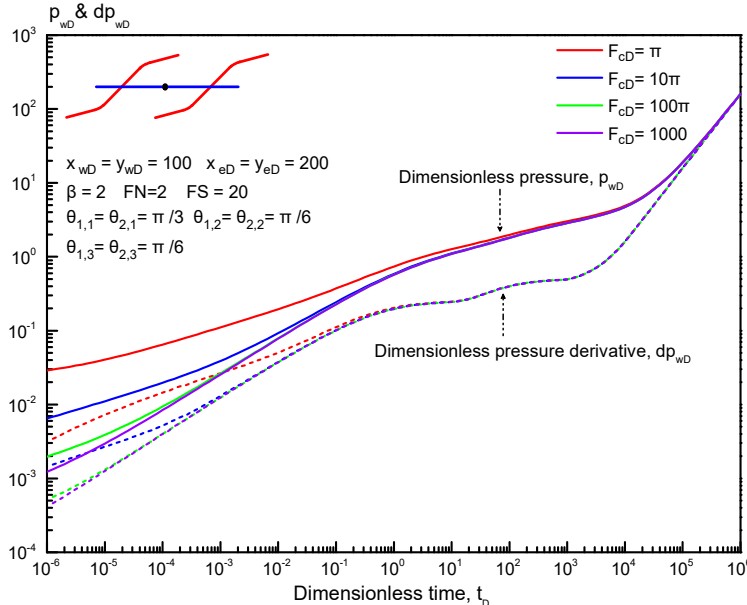

**Figure 15.** Effect of fracture conductivity on pressure and the pressure derivative.

Here PFA and RFA are kept constant and equal to 60° and 30°, respectively. As seen in Figure 15, the fracture conductivity affects the transient pressure significantly at the early-time period, mainly focusing on bilinear flow regime and first linear flow regime. The lower the fracture conductivity is, the earlier the bilinear flow regime appears. Moreover, when the fracture conductivity is low enough, the first linear flow regime becomes shorter and even tends to disappear. Meanwhile, the bilinear flow regime disappears and consequently the duration of the first linear flow lasts longer when fracture conductivity tends to be infinite ($F_{cD} > 100\pi$).

### 5.3.5. Effect of Dimensionless Fracture Spacing (FS) on Type Curves

Fracture spacing (FS) determines the intensity of flow interferences happened between the adjacent fractures [10], and can mask some typical flow behaviors. For example, the first radial flow regime disappears when FS is smaller than the fracture length. When FS is large enough, the flow interference occurs later, or each fracture flows independently and does not interfere with each other.

The influence of FS on type curves is presented in Figure 16 and it can be concluded that the difference among the various type curves is mainly concentrated on the duration of first radial flow regime and pseudo-radial flow regime. Smaller fracture spacing generates larger pressure depletion to maintain the constant production rate due to the strong flow interference between the adjacent fractures. Moreover, the larger the fracture spacing is, the later the pseudo radial flow regime appears and also the shorter this regime's duration is. For multiple fractured horizontal wells, the optimal fracture spacing mainly depends on the reorientation fracture number and the drainage area.

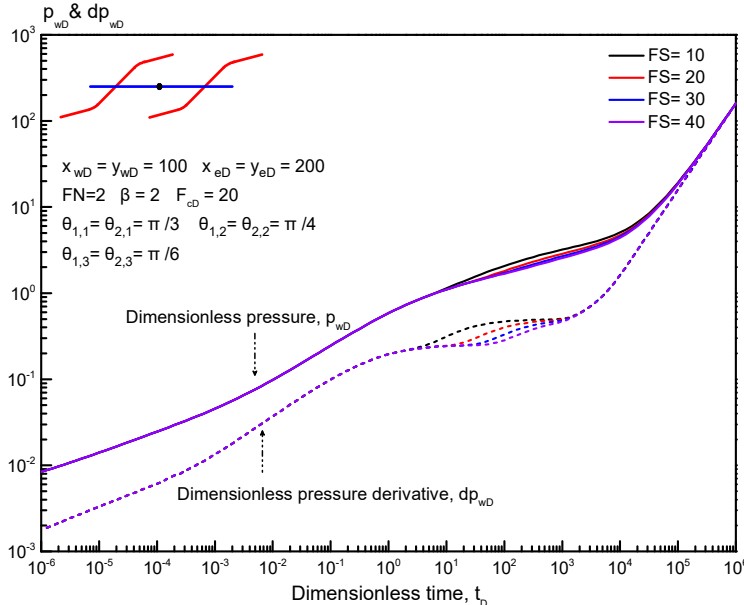

**Figure 16.** Effect of dimensionless fracture spacing on pressure and the pressure derivative.

### 5.3.6. Effect of Fracture Number (FN) on Type Curves

Fracture spacing between the adjacent fractures is assumed to be constant (FS = 20). The fracture number can control the area between fractures and the reservoir. Figure 17 presents the influence of fracture number (FN) on pressure transient. The increasing FN means an increasing fractured area when the adjacent fracture spacing is fixed. In general, the pressure drop decreases with the increase of the FN.

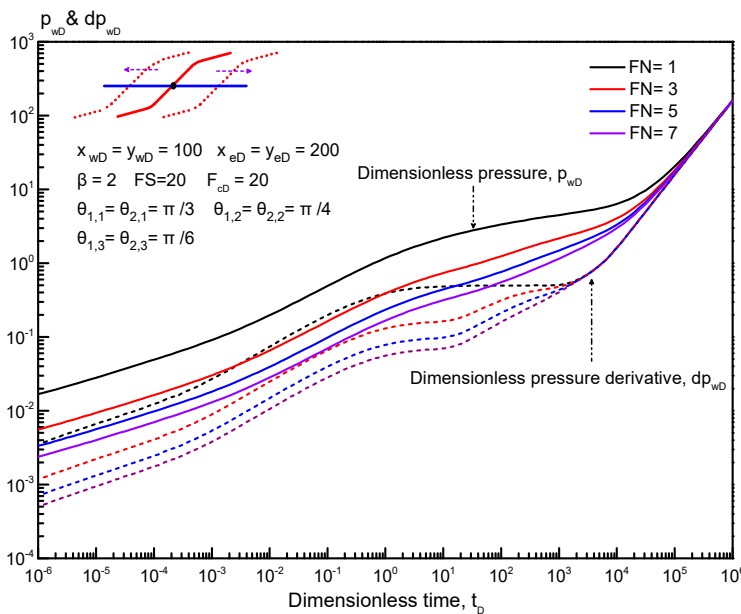

**Figure 17.** Effect of fracture number on pressure and the pressure derivative.

As seen in Figure 17, bilinear flow, first linear flow, first radial flow and second radial flow are all affected by reorientation fracture number. A large fracture number corresponds to a small pressure and derivative, suggesting that a smaller pressure drop is required to remain the same production rate. Consequently, the production is definitely enhanced by the increase in the fracture number, reflecting the advantage of multi-stage fractured horizontal wells. However, the degree of the decrease of the

pressure drop weakens when FN increases to some degree, indicating that the optimal reorientation fracture number needs to be demonstrated to maintain economic production.

### 5.3.7. Effect of Complex Fracture Configuration on Type Curves

Figure 18 investigates the influence of complex fracture configuration under large fracture spacing (FS = 20) on type curves.

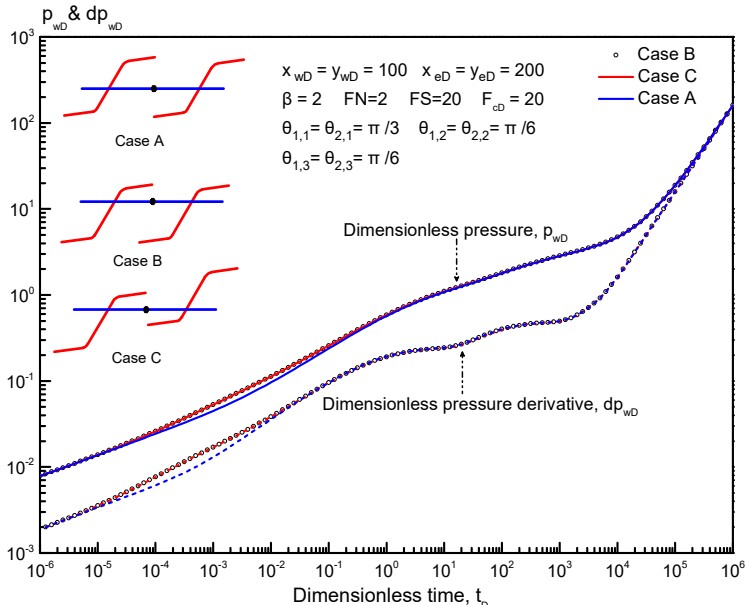

**Figure 18.** Effect of complex fracture pattern under large dimensionless fracture spacing on pressure and the pressure derivative (FS = 20).

For large fracture spacing, the influence of fracture configuration on type curves is largely focused on the transition flow regime ($10^{-5} \leq t_D \leq 10^{-1}$). Due to large fracture spacing, the difference between the effects of the staggered form of the reorientation fractures and the asymmetrical reorientation fractures on type curves can be neglected because each reorientation fracture behaves individually and does not interfere with each other. However, symmetrical reorientation fractures have higher production rates than the asymmetrical reorientation fractures when FS is large.

To determine the impact of fracture configuration on type curves under small fracture spacing (FS = 4), three typical cases are considered (Figure 19). Compared with the cases under large fracture spacing (Figure 18), high production rates are obtained when the reorientation fractures are staggered because the mutual dislocation of the reorientation fractures can reduce the flow interference between the adjacent reorientation fractures.

Further, the influence of different fracture configurations under the same fracture spacing (FS = 20) and fracture number (FN = 3) on type curves is illustrated in Figure 20. Case A depicts that the length of the outermost fractures are larger than that of the inner fractures, Case B represents that the length of the outermost fractures are smaller than that of the inner fractures, and Case C shows that each fracture in the system has the same length. Therefore, the equal fracture length model has a low pressure depletion under the same conditions, which suggests that during the hydraulic fracturing treatment the length of each fracture should be equal to maintain economic production (Figure 20).

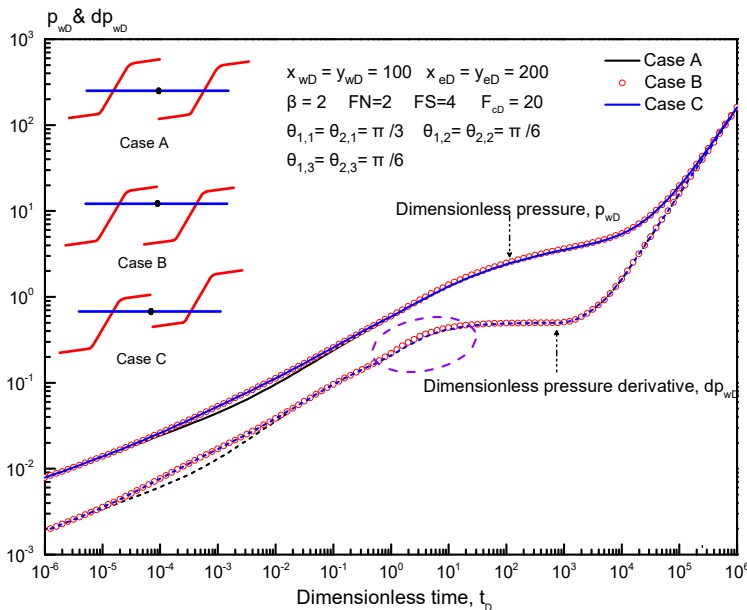

**Figure 19.** Effect of complex fracture configuration under small dimensionless fracture spacing on pressure and the pressure derivative (FS = 4).

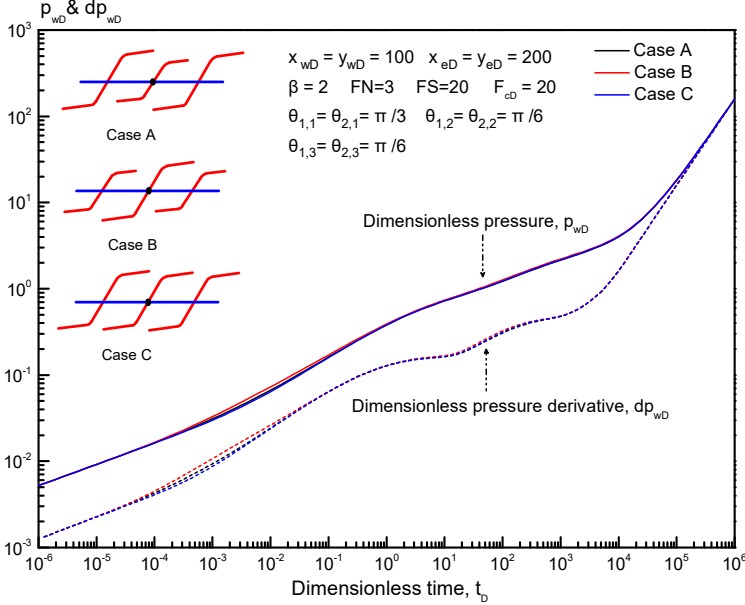

**Figure 20.** Effect of complex fracture configuration under different fracture distributions on pressure and pressure derivative (FN = 3).

## 6. Conclusions

On the basis of the new fracture equation, we present a semi-analytical solution to investigate the pressure behavior of multiple reorientation fractures along the horizontal well in an anisotropic tight reservoir. The main conclusions include the following:

(1) For a horizontal well perforated by multiple reorientation fractures with finite-conductivity in an anisotropic rectangular reservoir, six typical flow regimes can be found on type curves, including bilinear flow regime, first linear flow regime, first radial flow regime, second radial flow regime, pseudo-radial flow regime, and pseudo-steady-state flow regime. Meanwhile, the occurrence and duration of these typical regimes are determined by some significant parameters, such as

the permeability anisotropic factor, fracture reorientation, fracture spacing, fracture number, and fracture configuration.

(2) Fracture reorientation has an obvious effect on the production rate distribution along the fracture extension, particularly in the anisotropic tight reservoir. The production rate distribution curves for a horizontal well with three reorientation fractures in an anisotropic tight reservoir oscillate, and the outermost reorientation fracture has a larger production rate than the inner reorientation fracture due to a larger contact area with the reservoir.

(3) Fracture reorientation is one of the highlights of this work. For an anisotropic tight reservoir, the horizontal well should be deployed parallel to the direction of the principal permeability axis, and all reoriented fractures should be perpendicular to the direction of the horizontal well.

(4) The influence of permeability anisotropy on type curves is significant in all typical flow regimes except the late-time period. When horizontal wellbore is parallel to the principal permeability axis, first radial flow regime tends to disappear. However, when the horizontal wellbore is perpendicular to the principal permeability axis, the pseudo-radial flow regime changes into a compound linear flow regime.

(5) During the fracturing treatment, it is necessary to make the adjacent fractures stagger over each other and make the fracture spacing large to reduce the flow interference, and the length of each fracture in multiple fractured horizontal wells should remain the same to maintain economic production.

**Author Contributions:** All authors have contributed to this work. G.X. proposed the mathematical model and wrote the main manuscript text. As the corresponding author, M.W. made substantial contributions to the conception/design of the work. S.W., H.W., B.W., and J.C. contributed to the modeling, programming, results analysis, and discussion. J.W. mainly focuses on the modeling, programming of the manuscript.

**Funding:** This research was funded by the PetroChina Scientific Study and Technical Development Project (2017A-0906) and National Science and Technology Major Project of China (no. 2016ZX05046-003).

**Acknowledgments:** I am greatly indebted to my supervisor, Shuhong Wu, for her valuable instructions and suggestions on this work as well as her careful reading of the manuscript. In addition, I deeply appreciate the contribution to this work made in various ways by other co-authors.

**Conflicts of Interest:** The authors declare that they have no competing interests.

## Nomenclature

$x$ = Distance in the $x$-axis, m
$y$ = Distance in the $y$-axis, m
$x_e$ = Reservoir length in the $x$-axis, m
$y_e$ = Reservoir length in the $y$-axis, m
$h$ = Reservoir thickness, m
A = Drainage area, m$^2$
D = Distance between outermost fractures, m
$m_i$ = Total discrete number of all irregular curve fractures
$L_i$ = The length of the equivalent $i$-th planar fracture for the irregular curve fracture
$L_{fi,1}$ = Principal fracture length of the $i$-th reorientation fracture, m
$L_{fi,2}$, $L_{fi,3}$ = Reoriented fracture length of the $i$-th reorientation fracture, m
$\triangle l_i$ = length of the $i$-th fracture segment, $i$ =1, 2 ... $N_I$, $m$
$l$ = Fracture segment length, m
$L_R$ = Reference length, m
$w_f$ = Fracture width, m
$k_x$ = Permeability in the $x$-axis, $10^{-3} \mu m^2$
$k_y$ = Permeability in the $y$-axis, $10^{-3} \mu m^2$
$k_f$ = Fracture permeability, $10^{-3} \mu m^2$
$\theta_{i,1}$ = Principal fracture angle of the $i$-th reorientation fracture, rad

$\theta_{i,2}$, $\theta_{i,3}$ = Reoriented fracture angle of the *i*-th reorientation fracture, rad

$p_i$ = Initial reservoir pressure, MPa

$\phi$ = Reservoir porosity, fraction

$u$ = Fluid viscosity, mPa•s

$c_t$ = Total compressibility, 1/MPa

$q_{sc}$ = Wellbore flow rate, m$^3$/d

$q_{fw}$ = Flow rate of a reorientation fracture in the wellbore, m$^3$/d

$q_f$ = Rate of per unit fracture length from reservoir, m$^2$/d

$p_D$ = Dimensionless pressure in real time domain

$\tilde{p}_D$ = Dimensionless pressure in Laplace domain

$\tilde{p}_{fD}$ = Dimensionless reoriented fracture pressure in Laplace domain

$\tilde{q}_{fwD}$ = Dimensionless flow rate of a reorientation fracture in the wellbore in the Laplace domain

$\tilde{q}_{fD}$ = Dimensionless flow rate of per unit fracture length from reservoir in the Laplace domain

$q_{fD}$ = Dimensionless flow rate of per unit fracture length from reservoir in the real time domain

$q_D$ = Dimensionless point source flux

$F_{cD}$ = Dimensionless fracture conductivity

$l_D$ = Dimensionless fracture segment length

$t_D$ = Dimensionless time

$x_D$ = Dimensionless distance in the *x*-axis

$y_D$ = Dimensionless distance in the *y*-axis

$dp_{wD}$ = Dimensionless pressure derivative

$x_{wD}$ = Dimensionless distance in the x-axis of the center of the reservoir

$y_{wD}$ = Dimensionless distance in the y-axis of the center of the reservoir

RF = Reorientation fracture

FR = The location where the fracture reoriented

$x_{eD}$ = Dimensionless reservoir length in the *x*-axis

$y_{eD}$ = Dimensionless reservoir width in the *y*-axis

$R_{Dk}$ = Dimensionless coefficient

$s$ = Dimensionless time variable in Laplace domain

$N_I$ = Total fracture segment

$N$ = Total fracture number

$\triangle l_{Di}$ = Dimensionless length of the *i*-th fracture segment, $i$ =1, 2 … $N_I$

$x_{Dmi}$ = Dimensionless coordinate along the fracture extension

$y_{Dk}$ = Dimensionless coordinate along the fracture extension

cosh = Hyperbolic cosine function

sinh = Hyperbolic sine function

## Appendix A  Simplified Algorithm for Discrete Equations of a Reorientation Fracture

The discretized form of the reorientation fracture is:

$$\tilde{p}_{wD} - \tilde{p}_{fDi} = \frac{2\pi}{F_{cD}}\left[ l_{Di}\sum_{j=1}^{N_I}\left(\tilde{q}_{fDj}\Delta l_{Dj}\right) - \frac{\Delta l_{Di}^2}{8}\tilde{q}_{fDi} - \sum_{j=1}^{i-1}\left(\frac{\Delta l_{Dj}}{2} + l_{Di} - \sum_{n=1}^{j}\Delta l_{Dn}\right)\Delta l_{Dj}\tilde{q}_{fDj}\right] \tag{A1}$$

For $i > j$, Equation (A1) can be rewritten as:

$$\tilde{p}_{wD} - \tilde{p}_{fDi} = \frac{2\pi}{F_{cD}}\left[\sum_{n=1}^{j}\Delta l_{Dn} - \frac{\Delta l_{Dj}}{2}\right]\Delta l_{Dj}\tilde{q}_{fDj} \tag{A2}$$

For $i = j$, Equation (A1) is given as:

$$\tilde{p}_{wD} - \tilde{p}_{fDi} = \frac{2\pi}{F_{cD}}\left[l_{Di}\Delta l_{Dj} - \frac{\Delta l_{Di}^2}{8}\right]\tilde{q}_{fDj} \tag{A3}$$

Otherwise, Equation (A1) can be expressed as:

$$\tilde{p}_{wD} - \tilde{p}_{fDi} = \frac{2\pi}{F_{cD}} l_{Di} \Delta l_{Dj} \tilde{q}_{fDj} \tag{A4}$$

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
