# Peer review of "Pressure Transient Performance for a Horizontal Well Intercepted by Multiple Reorientation Fractures in a Tight Reservoir"

_energies, doi:10.3390/en12224232_

Round 1

Reviewer 1 Report

Dear Editor,

I was pleased to read this manuscript. The overall quality of the work is good. The paper is well organized and well written. This is a very interesting study that enables to “look” into the horizontal fractures, however as I’m not a specialist of this subject, I am not able to judge especially the physical model. Although it was rather difficult to understand and follow all the equations and steps, the overall picture and objective of this study are clear.

I have a couple of remarks.

- A comment is due to the application of this method to a real reservoir. It would be appropriate to apply this method to a real drilling case and include it in the discussion part.

- This application is refers to a tight reservoir. Is it valid in all sedimentary rocks? If and how the mineralogical composition and stress regime can affect the performance of the reservoir and thus your equations and physical model? Otherwise, this work remains only an excellent mathematical exercise.

- I always suggest to specify what kind of fractures are: hydraulic fractures or natural fractures. The term fractures can cause some confusion. Natural fractures are characterized by a complicated network. They can intersect each other and have completely different physical properties. Their physical properties depend on the type of rock they pass through, the fluids, whether they are open or closed, and the geometry. Natural fractures are not always equidistant. Hydraulic fractures, on the other hand, have different characteristics compared to natural fractures. Therefore, in my opinion, it is important to always specify the type of fracture.

Minor comments:

Line 69: However, with instead of However, With

Line 71: I suggest to be more precise about "soft" and "shallow" reservoirs.
soft in term of lithology? i.e. clay? shallow i.e. specify the depth range

Line 88: add a space between 2. and physical Model

Figure1: please add in the legend of the figure also that the angle q is the fracture angle and L is the length of the fracture.

Equations 1, 2, and 3: Please explain here the meaning of all parameters to avoid to the reader to check all time the nomenclature.

Discussion part: I strongly suggest to show an application of this theory to a real drilling case study.

Line 244: after Bilinear flow: in this regime (no capital letter)

Line 253: after linear flow: for the (no capital letter)

Line 267: after radial flow: after (no capital letter)

Line 273: after radial flow: in this regime (no capital letter)

Line 277: after stady-state flow: a pseudo (no capital letter)

Line 283: Figs instead of Fig

Line 28: in Fig.8 and 9, please modify with Figs. 8 and 9

Line 286: Table caption, please write Figs. 8-9

Line 303: add a space between reservoir and (

Line 346: please add a space between Fig. 13 and (, and also between Fig. 14 and (

Line 350: please add a space between in Fig.13

Line 396: please add a space between number and (FN)

Reviewer 2 Report

This paper gives comprehensive and valuable insight into the horizontal well fracturing techniques. The paper reads well. Major and minor revision notes are given in the attached file.

The major revision request is for the authors to clearly distinguish the scientific contribution of this paper when compared to the paper

Shuhong Wu, Guoqiang Xing, Yudong Cui, Baohua Wang, Mingyu Shi, Mingxian Wang, A semi-analytical model for pressure transient analysis of hydraulic reorientation fracture in an anisotropic reservoir, Journal of Petroleum Science and Engineering 179 (2019) 228–243.

The mentioned paper (marked reference number 21) was signed by almost the same group of authors. For instance, the semi-analytical method has already been given in that previous paper and shouldn’t be given as a novelty in the current paper but only the new elements should be emphasized.

Sincerely,

Author Response

Plsease see the attachment.

Reviewer 3 Report

The paper is well organized and overall well written. However, some more details should be given to help the readability. Moreover, figures should be improved.

More in details:

1) A brief introduction to fracture reorientation and a discussion of the difference between the current work should be given.

2) The description of the physical model and the corresponding figures should be clarified. As an example, does the blue line in fig. 1 represent the horizontal well or a principal fracture? What does the yellow point represent? A 3D image, like the one used by the authors in ref. 21, could help.

3) The differences between the previous work of the authors (ref.21) and the physical model of this work should be underlined and discussed in the introduction.

4) How the rates depicted in fig. 8 and 9 were calculated must be clarified.

5) The description (page 9-10) and the schematic (fig.7) of flow regimes for horizontal well with two reorientation fractures is not completely convincing. Why the second radial flow (fig.7d) should be characterized by a slope of 0.36 (line 269, pag. 9)? Being radial, it should be characterized by a slope 0. Looking at the figure 6 and the schematic (fig.7), I would consider it simply as a transition to the "Pseudo radial flow" (fig.7e), which correctly exhibit a slope 0. Concerning the Pseudo-steady flow, the image of fig.7f is not appropriate because it shows a radial flow not significantly different from fig.7e while the boundary effect is not depicted.

6) Eq. 3: Which length has been chosen as the reference length? (ex. fracture length, well length, ...?)

7) Fig. 5: Visual legend explaining A/D2 is hardly understandable. Add A, D2, LR to the nomenclature list. Moreover, the circle marker of the SPE legend is hardly visible.

8) Fig. 6: What do the circles in black represent? There is no legend of it. Moreover, review flow regimes identification (see point 5)

9) Fig. 8-Fig.9: these figures are not really effective because the meaning of x axes is not clear and the difference between fracture reorientation (FR) and reorientation fracture (RF) must be clarified.

10) In fig. 10 and fig. 11 some colors (blue and purple) are hardly distinguishable.

11) Fig. 12: the symbol of the reoriented fracture angle is missing in the legend.

12) Nomenclature is incomplete (ex. A, D, LR, FR, RF, xwD).

Round 2

Reviewer 2 Report

Proposed comments and requirements are fully met by authors and therefore I recommend publication of the manuscript.